# Diagnosing Infectious Diseases in Poultry Requires a Holistic Approach: A Review

Dieter Liebhart, Ivana Bilic, Beatrice Grafl, Claudia Hess and Michael Hess *

Clinic for Poultry and Fish Medicine, University of Veterinary Medicine Vienna, Veterinaerplatz 1, 1210 Vienna, Austria; dieter.liebhart@vetmeduni.ac.at (D.L.); ivana.bilic@vetmeduni.ac.at (I.B.); beatrice.grafl@vetmeduni.ac.at (B.G.); claudia.hess@vetmeduni.ac.at (C.H.)
* Correspondence: michael.hess@vetmeduni.ac.at

**Abstract:** Controlling infectious diseases is vital for poultry health and diagnostic methods are an indispensable feature to resolve disease etiologies and the impact of infectious agents on the host. Although the basic principles of disease diagnostics have not changed, the spectrum of poultry diseases constantly expanded, with the identification of new pathogens and improved knowledge on epidemiology and disease pathogenesis. In parallel, new technologies have been devised to identify and characterize infectious agents, but classical methods remain crucial, especially the isolation of pathogens and their further characterization in functional assays and studies. This review aims to highlight certain aspects of diagnosing infectious poultry pathogens, from the farm via the diagnostic laboratory and back, in order to close the circle. By this, the current knowledge will be summarized and future developments will be discussed in the context of applied state-of-the-art techniques. Overall, a common challenge is the increasing demand for infrastructure, skills and expertise. Divided into separate chapters, reflecting different disciplines, daily work implies the need to closely link technologies and human expertise in order to improve bird health, the production economy and to implement future intervention strategies for disease prevention.

**Keywords:** chicken; flock health; infectious diseases; virology; molecular biology; bacteriology; parasitology; serology

## 1. Introduction

Infectious diseases pose a constant risk on poultry health and production, with substantial consequences on welfare and economy. This is addressed in various book chapters focusing on infectious diseases, together with numerous review articles. Likewise, dedicated books and research articles suggest specific diagnostic techniques to diagnose infectious diseases in poultry.

The first successful example combining pathogen detection with initial characterization followed by the implementation of a screening program was the control of Pullorum disease. The concentrated action of different stakeholders paved the way for the National Poultry Improvement Plan (NPIP) in the USA and is a role model for successful cooperation between scientists, government and industry as reviewed in detail by Schat et al. [1]. Whereas some countries successfully eradicated the disease, others still rely on vaccination, which was developed in the mid of last century [2]. Interrupting vertical transmission as a main route of pathogen spread was also successful for eradication of avian leukosis J virus from primary broiler breeder flocks in some countries [3], again highlighting the close interaction of stakeholders. However, eradication is much less successful on a broad scale and poultry health is confronted with old and new diseases, sometimes summarized as (re-)emergent, today. As farms and flocks become bigger, economic losses of any disease outbreak are constantly increasing, although detailed figures, especially for endemic diseases, are very often not available [4]. Furthermore, in some areas, ethical

concerns about husbandry (cage vs. free range), management practices or breed of birds (e.g., fast- vs. slow-growing broilers) influence the epidemiology of certain pathogens and/or disease prevalence. Finally, despite the fact that backyard poultry is very often not covered by legislation which is applied to commercial poultry, diagnostic procedures should be performed to clarify disease etiology considering the importance of birds raised under loose biosecurity for pathogen spread [5,6].

Overall, in most cases, diagnosing infectious diseases follows an established procedure, starting at the farm with the collection of data on disease history. This is frequently combined with initial post mortems. Further investigations require a laboratory environment to proceed with different techniques and disciplines. This review aims to deliver an overview of standard procedures applied to diagnose infectious poultry diseases and to highlight new developments keeping in mind that each chapter requires a review on its own.

## 2. Diagnostic Activities on Farm

In poultry medicine, the diagnostic process originally shifted from the traditional veterinarian approach centered on individual animals to the health assessment of entire flocks. Flocks are commonly classified as "healthy" if they perform according to their genetic potential and are considered free from clinical disease. On-farm, diagnostic activities comprise routine sampling and investigations in line with health control programs; nationally and/or internationally adopted control programs for certain *Mycoplasma* and *Salmonella* species represent examples of paramount importance [7,8]. Samples may be investigated immediately on site (e.g., rapid antigen test for avian influenza) or sent for further processing to a laboratory (e.g., ELISA and PCR). Field veterinarians further implement diagnostic surveillance in order to provide epidemiological data for flock management purposes. The periodical collection of samples (e.g., feces, serum samples, and swabs from mucosal surfaces) is primarily used to confirm the infection (free) status of a flock or to monitor vaccine response. Altogether, generated data facilitate objective judgment and decision making in order to optimize flock health and production.

In the field, diagnostic procedures are initiated as soon as flock health is compromised, using morbidity and/or mortality as initial indicators. In such a scenario, investigations start with the compilation of a case history pertaining to relevant flock, management and infection/disease characteristics (e.g., bird type and origin, age, routine medications, vaccination program, previous diseases, husbandry system, standard operation procedures such as feeding and watering systems, ventilation, lighting program, hygiene and biosecurity processes, production parameters, morbidity and mortality data, duration of signs/problems and epidemiological links to other production sites). On the farm, diagnostics starts with the clinical examination of flocks, individual birds in various stages of the disease and their products (e.g., feces and eggs) by experienced poultry workers and veterinarians thoroughly familiar with the appearance of a healthy flock and the environment. Clinical examinations are time consuming and labor intensive and can regrettably fail to detect diseases; especially subclinical diseases can be challenging to be accurately diagnosed. The manifestation of an infectious disease can vary from subclinical to severe clinical illness, depending on various etiological factors and influences such as the causative agent, host and/or environment altogether complicating diagnosis [9]. Clinical signs comprise non-specific, general signs (e.g., apathy, ruffled feathers, and inappetence), which can be associated with a wide range of diseases, often together with more specific signs indicative of a certain disorder (e.g., enteric, respiratory, and neurologic) or even pathognomonic for a specific disease (e.g., histomonosis in turkeys) [10]. Diagnostic procedures continue with post mortem investigations, on farm or in the laboratory, which serve to identify gross pathologic changes in organs and tissues in order to further specify a tentative cause of impaired performance and clinical signs. For instance, high mortality in a turkey flock coinciding with sulfur-colored droppings paired with necropsy results of typhlitis and hepatitis indicates an infection with *Histomonas meleagridis*; a presumptive

diagnose, which might be confirmed by histology or even molecular diagnostics [11]. In the absence of mortality, euthanasia of sick birds might become a necessity to gain samples for further investigations. Altogether, a comprehensive case history together with the accurate assessment of clinical signs and thorough post mortem investigations narrows the range of presumptive diagnoses. This provides the basis to select appropriate laboratory methods as highlighted below in the outlook.

At the beginning, many infectious poultry diseases can be characterized by shortfalls in performance [12]. Those might be present long before clinical signs occur or even if clinical signs remain absent at all in infected flocks [4]. Consequently, production parameters (e.g., weight gain, feed-conversion rate, and egg production) of flocks are assessed and compared with established breed and/or company standards to detect anomalies. Very often production parameters are only available with certain delay (e.g., hatchability, chick quality or slaughter results) and can only be analyzed retrospectively with the purpose to enhance understanding of a diseases' impact and the implementation of corrective and preventive actions. Thus, financial losses due to subclinical disease outbreaks of necrotic enteritis have been calculated [13] and production losses as a result of emerging diseases have been detailed [14,15]. Similarly, the outcome and effectiveness of treatment plans have been evaluated analyzing historical flock data (e.g., coccidiosis) [16]. For diagnostic purposes, prompt evaluation of production efficiency and quality is essential. Modern poultry farming already uses precision livestock farming (PLF) technologies such as environmental sensors or platform scales to measure, predict and analyze various factors related to production in real time directly from within poultry houses [17]. So far, this is used mainly to optimize systems for feeding and drinking or to regulate environmental settings (heating and ventilation). Many characteristics of poultry production (e.g., closed housing system and large-scale integrations) provide a good opportunity to adopt PLF strategies to assess bird health and for early disease detection. Recently, modern technologic advancements utilizing video surveillance, audio recordings and/or wearable sensors have focused on behavior and behavioral changes in individual birds and entire flocks [18,19]. Thereby, bird activity, posture and/or vocalizations have been investigated in order to automate the detection of (early) infections and to identify disease outbreaks such as avian influenza or Newcastle Disease (Table 1). Likewise, fecal changes indicative of coccidiosis or salmonellosis have been assessed by imaging technologies to facilitate AI-based diagnostic services [20].

It seems obvious that sensor- or computer-based, automated systems will be part of the future of disease diagnostics, both in the field and in laboratories. However, the commonly rural location of farms might challenge field application of new diagnostic technologies and devices, which generally require a stable power supply and internet connection to function and provide data in real time [21]. Prospectively, as farm-generated data gains importance in poultry production and management, data security must be considered carefully to maintain company safety and integrity. Finally, the use of automated monitoring systems in intensive poultry production with a possible effect on the human–animal relationship may raise ethical concerns related to the objectification of animals [22].

**Table 1.** Precision livestock farming research focusing on infectious agents and related poultry diseases.

| Disease/Pathogen | Method (Variables Measured) | Type of Study * | Reference |
|---|---|---|---|
| Avian influenza | wearable sensor (body temperature) | experimental setting | [23] |
| Avian influenza | wearable sensor (activity, body temperature) | experimental setting | [24] |
| Avian influenza + Infectious bronchitis + Newcastle disease | sound analysis (vocalizations) | experimental setting | [25] |
| Avian influenza | imaging (posture) | experimental setting | [26] |
| Avian influenza | sound analysis (vocalizations) | experimental setting | [27] |
| Avian influenza | sound analysis (vocalizations) | experimental setting | [28] |

**Table 1.** *Cont.*

| Disease/Pathogen | Method (Variables Measured) | Type of Study * | Reference |
|---|---|---|---|
| Avian influenza | Imaging (thermal images) | experimental setting | [29] |
| *Campylobacter jejuni* | imaging (flock movement—optical flow) | dataset from broiler buildings | [30] |
| *Clostridium perfringens* | sound analysis (vocalizations) | experimental setting | [31] |
| Coccidiosis | sensor (volatile organic compounds) | experimental setting + broiler building | [32] |
| Coccidiosis | sensor (volatile organic compounds) | dataset from broiler buildings | [33] |
| Coccidiosis + *Salmonella* spp. | imaging (feces) | dataset of images | [20] |
| Ektoparasites | wearable sensor (activity) | dataset from poultry building | [34] |
| Infectious bronchitis | sound analysis (rales) | experimental setting | [35] |
| Infectious bronchitis | sound analysis (rales) | experimental setting | [36] |
| Infectious bronchitis + Newcastle disease | sound analysis (vocalizations) | experimental setting | [37] |
| Newcastle disease | sound analysis (sneezes) | experimental setting | [38] |
| Newcastle disease | imaging (posture and mobility) | experimental setting | [39] |
| Newcastle disease | sound analysis (vocalizations) | experimental setting | [40] |
| Non-specific, clinical signs | imaging (feces) | dataset of images | [41] |
| Non-specific, clinical signs | imaging and sound analysis | dataset of audio samples | [42] |
| Non-specific, clinical signs | imaging (feces) | dataset from broiler building | [43] |
| Non-specific, clinical signs | imaging (head motion, appearance) | experimental setting | [44] |
| Non-specific, clinical signs | imaging (posture, appearance) | experimental setting | [45] |
| Non-specific, clinical signs | sound analysis (abnormal respiratory sounds) | dataset from broiler building | [46] |
| Non-specific, clinical signs | imaging (posture, appearance) | dataset of images | [47] |
| *Pasteurella* spp. | imaging (thermal images) | experimental setting | [48] |

* All studies were perfomed with chickens, except Noh et al. [29] who also used ducks.

## 3. Bacteriology

In recent years, the extension of diagnostic tools in clinical microbiology laboratories had a major impact on bacteriological investigations. In general, a certain trend to introduce molecular techniques and proteomics that complement or substitute classical bacteriology can be observed [49]. The following chapter intends to highlight the value of classical bacteriology but also the contribution of new techniques to broaden the knowledge on relevant bacteria in poultry medicine.

Very often, the clinical picture and post mortem findings cannot be attributed to a specific bacteria or fungi. Therefore, one of the main goals of bacteriological and fungal examination in diagnostics is the isolation of the pathogen. In addition to disease diagnosis, culturing of bacteria is also obligatory for some surveillance programs, such as that of *Salmonella*, which are based on the isolation of live bacteria [50,51].

For culturing, material can be taken directly from affected organs or the environment. Either the streak plate procedure with subsequent dilutions to gain single colonies is applied or material will firstly be transferred into broth cultures and afterwards streaked out on solid media [52]. The culture media widely used in diagnostic laboratories have to be supplemented with diverse nutrients and components to enable or facilitate the growth of particular bacteria/fungi. To isolate fastidious bacteria blood agar containing different concentrations of sheep/horse blood is used which is also suited to assess the hemolytic ability [53]. MacConkey agar supplemented with bile salts and crystal violet inhibiting the growth of most Gram-positive bacteria is a well-established media for the isolation

of Gram-negative *Enterobacterales* [54]. More specific media, such as modified semisolid Rappaport Vassiliadis agar (MSRV) and modified charcoal-cefoperazone-deoxycholate agar (mCCDA) containing specific components and antibiotics, are used to isolate *Salmonella* species and thermophilic *Campylobacter*, respectively [51,55]. For cultivation of fungi and yeasts different variations of Sabouraud-glucose agars are mainly used [56]. In general, bacteria and fungi of interest have their growth optimum between 37.0 °C and 41.5 °C [52]. Some bacterial pathogens have special atmosphere requirements. For example, strict microaerobic conditions are needed for the cultivation of *Avibacterium paragallinarum* [57]. In contrary, most *Brachyspira* and *Clostridium* species will strictly grow under anaerobic conditions only [58,59]. Some bacteria are difficult to cultivate, relying on complex media and longer incubation time to achieve sufficient growth, e.g., *Mycoplasma* species, *Avibacterium paragallinarum*, *Campylobacter hepaticus* and *Mycobacterium avium* [57,60–62]. In addition to suitable media and cultivation conditions, classical bacteriological cultivation is also limited by the appearance of viable but non-culturable (VBNC) bacteria which are known to develop under unfavorable environmental conditions reported for *Campylobacter jejuni* and *Escherichia coli* [63,64].

The classical cultivation techniques do not only provide information if pure or mixed bacterial cultures are present, it also enables further typing, the application of antimicrobial susceptibility tests, production of autogenous vaccines and execution of functional studies. Traditional typing methods comprise the determination of colony morphology, hemolysis, staining and phenotypic identification tests. Based on colony morphology (size, shape, color, margin, surface) unusual growth variants of bacterial species can be detected as previously shown for *Avibacterium paragallinarum* and *Ornithobacterium rhinotracheale* [65,66]. Hemolysis is used to group bacteria into non-hemolytic such as *Pasteurella multocida*, partial-hemolytic (α-hemolysis) such as *Erysipelothrix rhusiopathiae* and complete-hemolytic (ß-hemolysis) such as *Staphylococcus aureus*. Most often used staining method is the Gram-stain differentiating bacteria by chemical and physical properties of their cell walls into Gram-positive and Gram-negative. The Ziehl-Neelsen stain is used to identify acid-fast organisms such as *Mycobacterium avium*, and the lactophenol-blue stain is applied to illustrate fungi and yeasts. Phenotypic identification tests comprise diverse biochemical reactions for which commercial systems are available, e.g., analytical profile index (API) systems [67]. Such tests can also be applied to discriminate subspecies. In *Pasteurella multocida*, the classification of the three subspecies, subsp. *multocida*, *septica*, or *gallicida*, is based on the ability to ferment sorbitol and dulcitol [68]. However, these biochemical methods are time consuming and resource intensive. Furthermore, variable characteristics among members of the same species or the lack to ferment most of the carbohydrates results in a heterogeneous or insufficient outcome of testing as shown for *Gallibacterium anatis* or *Riemerella anatipestifer* [69,70]. Worth to mention that also commercially available phenotypic databases sometimes lack updated bacterial classification data hindering proper and reliable identification as shown for *Ornithobacterium rhinotracheale* [71]. Furthermore, serotyping schemes are widely used based on antibodies raised against certain bacterial components such as capsules, somatic structures or flagella. Autoagglutination, cross-reactions and non-typeable bacterial strains are regularly found, altogether limiting the approach [71–74].

Live bacteria are also needed to determine antibiotic resistance profiles in order to implement a targeted treatment and to collect data for long-term assessment of antibiotic resistance [75–79]. Most common in routine diagnostics is the application of the disc diffusion method [80], which is still the golden standard to test antimicrobial susceptibility. The use of the broth microdilution method, for which commercially available semi-automated systems are available, is also popular [81]. The evaluation of the outcomes from such testing is based on standard documents, e.g., EUCAST [82] and CLSI [83].

Due to limitations in cultivating and typing, molecular techniques and proteomics has become more popular in recent years in routine bacteriological diagnostics [84,85]. For example, different PCR assays are widely used for the detection of *Mycoplasma gallisepticum*

and *Mycoplasma synoviae* [86], *Chlamydia* spp. [87], *Gallibacterium anatis* [88], *Avibacterium paragallinarum* [89] or *Salmonella* spp. [90]. Specific PCR assays are also applied for species identification and further typing of isolates, e.g., defining virulence-associated genes in *Escherichia coli* or the Type C neurotoxin (BoNTC) gene in *Clostridium botulinum* [91,92]. In addition to, sequencing based on 16S and 23S ribosomal RNA (rRNA) regions as well as next-generation sequencing (NGS)/whole-genome sequencing (WGS) are increasingly applied. Sequence analysis of rRNA has been used for example for species identification of *Campylobacter jejuni* and *Campylobacter coli* [93], or *Avibacterium paragallinarum* [94]. NGS/WGS proved valuable in gaining more data regarding characterization of *Morganella morganii* [95], *Ornithobacterium rhinotracheale* [96], *E. coli* [97] and *Salmonella* Infantis [98]. These methods can also be applied for direct identification of bacterial communities as shown for samples from poultry carcasses in slaughterhouses [99]. A new area of application for such culture-independent assays is the screening of gut microbiota, but also environmental samples for antibiotic resistance genes to obtain a better understanding of the antibiotic resistome [100–104].

In addition to genetic methods, proteomics found its way into routine diagnostics mainly due to the implementation of matrix-assisted laser desorption/ionization time-of-flight mass spectrometry (MALDI-TOF MS), which is based on ribosomal protein profiles obtained from bacterial isolates [84,105]. The availability of benchtop instruments makes it easier to develop in-house databases to be used in addition to commercially available ones. With this, a broad range of bacteria relevant in poultry medicine can successfully be identified to the species level, as shown for *Gallibacterium* [106], *Riemerella* [107], *Mycoplasma* [108], and *Aspergillus* [109,110], whereas limitations for others such as *Avibacterium* were reported [111]. Investigations beyond the species level were able to define clonal lineages for *Gallibacterium anatis* [112] or to discriminate environmental source specific isolates of *Escherichia coli* [113]. MALDI-TOF MS also has several applications to detect specific feature of antibiotic resistance, e.g., enzymatic activity (carbapenemases) or direct analysis of bacterial extracts (vancomycin-resistant enterococci) [114], with the advantage that results are available within a few hours.

Both molecular methods and proteomics are suited to determine phylogenetic relationships of isolates [112,115,116]. This feature proved of value in surveillance programs as shown for *Campylobacter* and *Salmonella* spp. [117,118]. It also highlights the importance of cultivating bacteria, which, in our opinion, remains irreplaceable despite modern technologies. It is crucial to perform animal trials and to investigate host–pathogen interactions. Live and well-defined bacteria also provide the basis for the production of autogenous vaccines, an important aspect of disease prevention [119–122].

However, with the help of more advanced technologies such as PCR and MALDI-TOF MS, the direct detection of pathogens, especially those which are difficult to cultivate, as well as routine identification of bacteria and fungi have become more rapid and easier during the last decade. Furthermore, NGS is currently transitioning from research to diagnostics, becoming more common in diagnostic laboratories. In regard to antimicrobial susceptibility testing, phenotypic methods will still remain crucial and its replacement by other techniques in the near future will not be the case. Finally, serological tests implemented in the field will still remain valuable tools for official monitoring programs as outlined below in Section 7.

## 4. Parasitology

Diagnoses of poultry parasites depends particularly on macroscopical and microscopical examination [123]. Ectoparasites include insects and arachnids that can be observed during clinical examination and necropsy on the feathers or the skin of infected poultry. Changes in the integument and its structures display a first suspicion that can be further specified by careful examination with the naked eye [124]. However, to determine the level of infestation, traps positioned in various areas of the poultry house are needed [125]. Internal parasites of poultry can colonize different organs or the blood [126,127].

The growing trend of raising laying hens in alternative housing systems with less biosecurity favors the prevalence of parasites [128,129]. Hence, Jung et al. [130] showed that in such systems, the majority of flocks excreted more than 200 ascarid eggs per gram feces. Accurate necropsy can reveal the presence of larvae and adult stages of helminths; however, eggs and protozoan parasites have to be identified by microscopic examination. For intestinal parasites, scrapings of the mucosa, intestinal content or feces are used and assessed for the presence of eggs. The identification of parasites and/or eggs can be achieved considering their morphological characteristics and depends on the applied technique and the experience of the investigator [131]. The applied diagnostic method is of high importance. The flotation method in combination with post mortem investigations is recommended for the detection of intestinal helminths to reduce false negatives [132]. Recently, fecal egg count techniques have been evaluated and it was demonstrated that McMaster is more accurate but less precise than mini-FLOTAC, underlining the need to evaluate diagnostic techniques [133]. Flotation and enumeration can also be applied for intestinal protozoans such as *Eimeria* spp. [134]. Other intestinal protozoans, such as flagellates, can be diagnosed by microscopic detection from cecal content or following isolation in culture medium but this requires fresh sample material [10]. Alternatively, molecular detection tools such as PCR, immunohistochemistry and in situ hybridization, as comprehensively described in the respective chapter of this review, have been developed for a sensitive and specific diagnosis of parasitic diseases of poultry.

In addition to conventional techniques, PCR can be applied to increase sensitivity and specificity to detect parasites with the advantage of combing it with typing as shown for *Eimeria* spp. [135,136]. In addition to tissue samples, environmental samples might be screened by PCR for parasite DNA even though clinical relevance seems less clear [137–139]. Furthermore, indirect detection by serology can be applied, which is helpful to generate results from high numbers of birds. Experimental ELISA systems were developed to determine antibodies against *Ascaridia galli* and *Histomonas meleagridis* [140–142]. Although there are a few research studies demonstrating the value of the assays, none of them are commercially available, limiting their use as monitoring tools.

## 5. Virology

Poultry is the host of numerous viruses, with some of them infecting chickens, turkeys, waterfowl, spreading to wild birds and even humans. The most important examples are avian influenza viruses with a broad host range and substantial variation in disease pathogenesis depending on host–virus interaction. Diagnosis of viral infections is a paradigm of a multifaceted approach as PCR has become a method of choice enabling fast detection of the nucleic acid (see below Section 6). However, to resolve functional aspects of an infection or to gain antigen for vaccine production, very often isolation is needed. However, multiplication of viruses relies on a certain substrate but the lack of broadly susceptible cell lines is very obvious in a poultry diagnostic laboratory. In general, initial isolation of a virus is somehow different to adaptation and propagation of a pathogen already available. In this context, poultry virology is still dominated by embryonated specific pathogen-free (SPF) chicken eggs and cells obtained from those embryos or hatched SPF birds. In a diagnostic laboratory, SPF eggs are very often at hand and the broad range of susceptibility is a substantial surplus, albeit additional work is needed to characterize and type the isolate which might be considered as a second step.

The first epithelial cell line from fowl was established from the liver of carcinogen treated chickens [143]. In one of the first studies, this hepatocellular carcinoma cell line (LMH) was capable of supporting multiplication of infectious laryngotracheitis virus (ILTV) although initial isolation was not reported [144]. Chicken astrovirus (CAstV) was successfully isolated on LMH cells, inducing a marked CPE after 3–5 passages [145,146]. In an earlier study, the susceptibility of the cell line to multiplicate Avian Nephritis Virus (ANV) and duck astroviruses (DAstV) was demonstrated as well. Isolation of DAstV-1 was described later on, again with CPE at the 5th passage [147,148]. The same number

of passages was reported to induce visible changes in LMH cells during isolation of goose astrovirus [149]. With the recent appearance of virulent fowl adenovirus (FAdV) serotype 4 in China, LMH cells have become attractive to substitute SPF embryos for virus isolation [150]. However, in a very comprehensive epidemiological study, out of 2210 PCR positive samples, only 877 FAdV isolates were obtained despite 6 blind passages on LMH cells [151]. Although the difference might be explained by the absence of live virus, it could also well be that some serotypes/strains are more easily isolated and/or LMH cells might be a less suitable substrate compared with primary cells and/or SPF embryos. Except the initial study about the growth of FAdV-9 on LMH cells, no additional serotypes are investigated in detail, although cells are widely used in recent studies on virulent FAdV-4 in China [152,153]. Isolation of avian reoviruses from chickens, turkeys, pheasants and guinea fowl was very successful on LMH cells with CPE already detected after 24 h [154]. Similarly, a recombinant ARV isolate from geese with high sequence homology to chicken isolates could be isolated using LMH cells [155]. A single report described the isolation of avian Hepatitis E Virus (aHEV) on LMH cells which contradicts an earlier report that cell to cell spread of the virus is impossible following transfection of LMH cells with an infectious aHEV clone [156,157].

Several cell lines were established from chemically induced fibrosarcoma of Japanese quails [158]. On one of those, QT-35 was able to induce a cytopathogenic effect by testing a broad range of avian viruses, although variations among virus strains were noticed [159]. Strain variations have become obvious as none of the tested adenoviruses induced a CPE but later on the attenuation of virulent FAdV-4 on QT-35 cells was reported [160]. QT-35 cells also support initial replication of ILTV but continuous passaging was not possible [144,159]. These quail cells also support the induction of a CPE by reticuloendotheliosis virus (REV) [161]. Being equipped with a high number of α2,3-gal receptors, avian influenza viruses from different birds grew best on QT-6 cells, compared with DF-1 and MDCK cells [162].

Contrary to those reports on multiplication of already isolated viruses, QT-35 cells were used to isolate an orthoreovirus from tendons of turkeys suffering from arthritis after several passages [163]. Similarly, avian metapneumovirus subtype C, either from commercial turkeys or wild Canada goose, was successfully isolated on QT-35 cells. Success for other aMPVs was not reported although multiplication was demonstrated [164–167].

A cell line with susceptibility for different viruses is the chick embryo related (CER) cell line, a mixture of chicken embryo fibroblasts and baby hamster kidney (BHK21) cells [168]. It was used to isolate avian metapneumovirus (aMPV) from turkeys and it was later shown to support growth of infectious bursal disease virus (IBDV) and infectious bronchitis virus [169–171].

The DF-1 cell line is a fibroblast cell line with the special feature of carrying no endogenous avian sarcoma and leucosis virus [172]. With this feature, the cell line is well suited for isolating avian leucosis viruses [173]. An infectious bronchitis virus that causes kidney lesions and respiratory signs can induce a CPE in DF-1 cells, but without being passaged that is different to the observation in QT-35 cells, in which passaging could be achieved albeit with low titers [159,174]. Similarly, higher titers were obtained with primary chicken embryo fibroblasts as compared to DF-1 cells, even though both cell culture systems multiplied IBDV and NDV [175].

Frequent studies describe the isolation of viruses infecting poultry on mammalian cell lines, especially kidney cells. Following initial isolation in tracheal organ culture and adaption of aMPV on Vero cells such cells were later also used for primary virus isolation [164,176–179]. Vero cells are also described to be susceptible for IB, following primary isolation in SPF embryos [180]. First successful isolation of avian group A rotaviruses in Rhesus monkey kidney (MA-104) cells was reported from feces of diseased turkeys [181]. The Madin–Darby canine kidney (MDCK) cell line is widely described for isolation and propagation of influenza A virus but QT-6 and DF-1 cells might be favorable for propagating isolates from poultry due to high presence of sialic acid α2,3-gal linked receptors [162,182]. However, routinely, SPF eggs remain a gold standard for isolating AIV

as indicated in the relevant chapter of the OIE manual [183]. Vero cells, DF-1, baby hamster kidney cells (BHK) and other vertebrate cell lines might be used to isolate Arboviruses (West Nile Virus, or Eastern Equine Encephalitis Virus) keeping in mind that some viruses, e.g., Turkey meningoencephalitis virus, rely on primary chicken cells [184].

Of high importance for avian virology are tumor cell lines established in the course of Marek's disease. One of those cell lines, MDTC-RP19, established from turkey B cells, can be used to isolate and propagate hemorrhagic enteritis virus of turkeys (Turkey adenovirus 3 (TAdV-3) [185]. Similarly, chicken anemia virus (CAV) can be isolated and propagated in a MDV transformed T-cell line (MDCC-MSB1) established from a tumor in chickens [186].

A spontaneously immortalized turkey turbinate cell line displayed good susceptibility for avian metapneumovirus with higher titers than in commonly used Vero cells, albeit with delayed replication [187]. No other viruses were tested on this cell line although quite some potential would be available considering the number of pathogens targeting the respiratory tract as mentioned by the authors.

Despite the ability of cell lines reported above there are severe limitations for certain viruses to be isolated and the best examples are members of the *Picornaviridae* for which isolation is restricted to host or chicken embryos. This includes avian encephalomyelitis virus (AEV) but also turkey hepatitis virus (THV), and duck hepatitis A virus (DHAV). Similarly, isolation of turkey coronavirus (TCoV), chicken parvovirus or proventricular necrosis virus (CPNV), a member of the *Birnaviridae*, rely on SPF embryos [188,189]. Different to avian group A rotaviruses which can be isolated in MA104 cells success to isolate members of groups D, F and G from poultry species is very much limited [190,191]. Except CAV, circoviruses from waterfowl such as ducks (DuCV) or geese (GoCV) are isolated in respective embryos which are also used for isolating parvoviruses. Similarly, for goose parvovirus (Derzy's disease) or Muscovy Duck parvovirus, both inducing hepatitis, isolation relies on embryos or primary cultures. A recently described goose pegivirus (GPgV), family *Flaviviridae*, was isolated in goose embryos and transferred to goose embryo fibroblast albeit with only a low increase in titer and no CPE was noticed [192].

In summary, it is very obvious that SPF eggs or primary cells are still an indispensable feature in a diagnostic laboratory. The susceptibility for numerous viruses is a clear surplus but the preparation is time consuming and costly. In addition, it needs a constant supply, and the procedures are difficult to standardize. Furthermore, there remain ethical concerns with regard to the use of primary cells and SPF embryos. However, it is difficult to believe that this situation will change in the future, since activities on established poultry cell lines are neglectable as no major achievements to establish cells lines from poultry species are reported in recent years.

## 6. Molecular Diagnostics

Molecular diagnostics comprises technologies identifying a disease or a pathogen by examining molecules, such as nucleic acids and proteins, in biological materials and environmental samples. In poultry medicine, molecular diagnostics is utilized to detect and analyze pathogen-specific DNA and/or RNA molecules. Although successful detection indicates the presence of the pathogen's nucleic acid, it does not provide information if the detected pathogen was alive. Compared to conventional virological, bacteriological and parasitological investigations, molecular diagnostics offers a fast and specific alternative to detect an infectious agent without requirement of a live pathogen. In this respect the application of molecular diagnostics is superior, because samples can be inactivated before shipping, which simplifies their transport to a diagnostic laboratory. A good example is the use of Flinders Technology Associates (FTA) cards®, an envelope-like covering made of filter paper treated with a patented chemical mix that lyses cells, denatures proteins and preserves nucleic acids. In this way, the applied sample remains suitable for molecular diagnostics without the risk of spreading an infectious agent [193]. Nonetheless, the possibility that some infectious material might remain on the FTA card argues for careful handling since some studies reported detection of viable viruses or bacteria after their application on the

FTA card [194,195]. In addition to use the intentionally inactivated material, the advantage of molecular diagnostics lies in the capacity to analyze a great variety of sample material such as all kinds of tissues and organs including formaldehyde-fixed paraffin-embedded (FFPE) material, body fluids, eggs, feathers, swabs and environmental samples such as dust, soil or litter.

In poultry medicine, the term molecular diagnostics generally considers various PCR-based methods. Even though strictly taken, methods such as in situ hybridization, immuno-staining or Enzyme-Linked Immunosorbent Assay (ELISA) also directly analyze molecules such as nucleic acids or proteins, they are considered as part of histology or serology and will be discussed therein. Due to easiness, speed and at the same time high specificity of PCR diagnostics detection of viral poultry pathogens is currently almost entirely based on this method [196]. In contrast to this, the detection of bacterial and parasitic pathogens is very often still carried out by classical bacteriology or parasitology. Exceptions are special situations in which the cultivation of the pathogen is either time consuming and/or complex such as for example the cultivation of *Mycoplasma* strains or impossible due the inherence of the starting material as FTA card or FFPE tissues [193,197]. Some molecular diagnostic assays are qualitative by nature assessing the presence or absence of the pathogen's nucleic acids. Both conventional and real-time PCR tests are employed, although the prevailing tendency goes toward the fluorescently labelled oligonucleotide probe-based real-time PCR tests. This is mainly due to their greater specificity provided by the presence of a sequence-specific oligonucleotide probe as compared to conventional PCR or SYBR Green-based real-time PCRs, which lack this feature [198]. Furthermore, the capacity to specifically detect different targets within a single reaction, so called multiplexing, provides an additional surplus. Multiplexing is readily employed in PCR tests using an internal positive control, in which pathogen's nucleic acid is detected in parallel to an independent nucleic acid target, either an endogenous internal control such as host-specific target or artificially added target DNA or RNA representing an exogenous internal control [199,200]. Inclusion of an internal control in a PCR assay is a requirement for assays accredited by ISO17025 to determine the success of the PCR within a given matrix [201]. Noteworthy is the implementation of multiplex PCR assays in i) detection of more than one pathogen in a course of a single PCR, or ii) for pathogen typing. Various assays for simultaneous detection of different pathogens have been described, such as protocols based on multiplex real-time PCR in detection of fowl poxvirus and reticuloenotheliosis virus [202,203] or detection of *Mycoplasma gallisepticum* and *Mycoplasma synoviae* [204] or conventional assays such as simultaneous detection of picorna-, astro- and caliciviruses [205] or Campylobacter, Arcobacter and Helicobacter species [206]. Similarly, diverse multiplex PCR assays are available for detecting and typing a pathogen within the same assay, and avian influenza viruses subjected to multiplex real-time PCR are a good example [207,208]. Similarly, rapid serotyping of *Erysipelothrix rhusiopathiae* by conventional multiplex PCR can be mentioned [209,210].

The multiplexing scheme is also included in the high-throughput methods using devices such as Genome Lab Gene Expression Profiler (GeXP) or Luminex in analysis of PCR products. The GeXP system uses capillary electrophoresis for fine separation of PCR products, detecting length differences of just few base pairs which enables the detection of up to 30 different targets [211]. The use of GeXP-based PCR methods was reported for both (i) concurrent detection of different pathogens and (ii) for pathogen typing [212–216]. Another application involving multiplexing is the alliance of PCR with the Luminex system, in which the detection of different targets is based on measuring beads that are coupled with target specific nucleic acids [217]. Similar to the GeXP system, bead-based multiplex assays are applicable for high throughput and can simultaneously analyze up to 50–500 different targets within a single sample depending on the instrument and assay applied. In diagnosing poultry infectious agents bead-based multiplex PCRs were used in (i) parallel detection of different pathogens and (ii) typing of pathogens [218–222]. The clear advantage of both GeXP-analyzer- and Luminex-based methods is the much higher number of targets that can be detected in

a single reaction as compared to real-time PCR or conventional PCR, which speeds up the diagnostics process. However, the high price of devices which are essential for such analyses (GeXP-analyzer or Luminex), challenges the justification for these methodologies, especially if it is not used at full capacity.

Specific application of PCR is a DIVA (differentiating infected from vaccinated animals)-PCR, a method used for screening a vaccinated flock for infections with a filed/virulent strain, as for example in Marek's disease virus (MDV), *Mycoplasma gallisepticum*, *Mycoplasma synoviae* or turkey meningoencephalitis virus [223–226]. In order to accomplish functionality, DIVA tests should be able to distinguish between a vaccine and a field strain even in situations when multiple strains are present within the sample. In some cases, such as MDV Rispens or CVI988 vaccine and virulent/oncogenic MDV strain, difference between the vaccine strain and wild-type/virulent strain is often minor, which introduces challenges in methodology. In the context of differentiating vaccinated and unvaccinated birds, method Mismatch Amplification Mutation Assay PCR (MAMA-PCR) has been applied for MDV, *Mycoplasma gallisepticum* and *Mycoplasma synoviae* [227–229]. The approach is based on the presence of a single nucleotide polymorphism (SNP) between vaccine and field/virulent strain, which is incorporated at 3′-end of one of the primers. The incorporated SNP causes mismatching of the primer with the template, which should ultimately result in the abolition of amplification. However, the stringency of this process is not high and very often one of the specific primers demonstrates a certain level of cross-reaction with the opposite template detecting the unwanted strain/type, such as CVI988 vaccine-specific primers detecting MDV oncogenic strains [227]. Therefore, a set of good controls for evaluating the MAMA test system is needed.

In addition to sole PCR-based assays, pathogen typing often includes a combination of conventional PCR with sequence analysis of the PCR product. The advantage of such an approach is a much broader detection range of strains/types/variants, as for viral pathogens with high diversity such as FAdV, infectious bronchitis virus (IBV), IBDV or Newcastle disease virus (NDV) [230–237]. Targets used for typing/classification are in general genes coding for surface proteins which display high variation among different strains/variants as compared to pathogen's proteins not exposed to the host. This characteristic is directly related to the immune response of the host and frequently also to an earlier classification method such as serotyping, patho-typing, or protecto-typing. Sequence variations often cluster in specific gene regions that are flanked by a more conserved sequence, the aspect used for designing the primers to amplify as many variants/strains as possible. Classification employs a phylogenetic analysis of obtained sequences integrating them with available data from reference strains.

In recent years the implementation of NGS and third-generation sequencing (TGS) in diagnosing infectious diseases in poultry has gained its importance, as rapid advancements in sequencing technologies, bioinformatics and computational tools have made these approaches broadly available [238]. Both methodologies enable random sequencing of all nucleic acids in the sample, detecting potentially all microbes and viruses. Such assumption-free procedure enables the discovery of (un)expected or novel pathogens (re)defining the etiology [239]. This is especially valuable in cases that have exhausted the available diagnostic analyses, as for example, in disclosure of a etiological agents of fulminating enteritis of French guinea fowl or hepatitis in pheasants, which were only possible with metagenomic NGS of samples from diseased birds [240,241]. Since metagenomic analyses produce vast amount of data, scrutiny of all available clinical, diagnostic and epidemiological results is required for accurate evaluation of the respective case. Metagenomic analyses are currently also very popular in epidemiological investigations of various health issues and frequently result in finding novel species [239,242].

However, proper study designs and functional knowledge about microbes and viruses in a certain host are a prerequisite for an accurate evaluation of microbial species involved in clinically relevant cases. The NGS or TGS of clinical samples have also been shown beneficial for detecting pathogens with high mutation rate such as RNA viruses, which due

to their error-prone genome replication and re-assortment of genome segments demonstrate increased genetic variation [243–245]. Since classical molecular detection using PCR methods is based on specific targets in pathogen's genome, introduction of new mutations might hinder their proper detection [246]. In addition to the fact that the application of NGS and TGS opens powerful modern strategies in molecular diagnostics, several drawbacks or bottlenecks limit these methodologies from being a tool of choice. Pathogen detection is more expensive and labor intensive compared with classical molecular diagnostics.

In particular, the generation of a large quantity of raw data and their complex analysis demand high-performance computers and extensive bioinformatics analysis [247]. Furthermore, untargeted sequencing of total DNA and/or RNA from clinical samples results in very low amount of microbes and especially viral nucleic acids as compared to the host cellular DNA and RNA; therefore, special sample preparation such as depletion of host nucleic acids and /or enrichment of viral DNA/RNA is frequently required [248–253]. However, considering the speed of the development of techniques, protocols and tools focused on NGS and TGS, one can foresee that these technologies will soon emerge as the tool of choice in molecular diagnostics.

## 7. Serology

After contact with an infectious agent, the poultry immune system frequently reacts with the production of antibodies. Typically, serology uses specific antigen–antibody reactions for the detection of bacteria, viruses or parasites that may be difficult to detect by other methods. In contrast to the isolation or detection of infectious agents, especially viruses, serological testing is fast and easy to perform with minimal laboratory requirements. Primarily, it focuses on the presence, absence or level of specific antibodies in the serum. In practice, antibodies can be detected in a number of body fluids such as egg, tears, saliva, and mucus secretions [254,255].

Many of the classical serological techniques (e.g., the agar gel precipitation test, plate agglutination tests (RSA), and hemagglutination inhibition tests (HI)) have been employed for decades; standardized reference sera and antigens for many poultry pathogens are commercially available worldwide. Considering the high flexibility, quick turnaround and the large sample size to be processed at a time, ELISA assays have become routine procedures for flock health monitoring and for poultry diagnostics alike, with commercial systems available for the majority of poultry pathogens.

In addition to direct pathogen detection described above in Sections 3 and 6, some national/international poultry health control programs for certain infections/diseases (*Mycoplasma gallisepticum*, *Mycoplasma meleagridis*, *Salmonella* Gallinarum/Pullorum, and avian influenza) employ sero-surveillance to provide evidence for circulating infections within poultry populations [256–259]. Office International des Epizooties (OIE) standard serological tests—RSA and ELISA—are recommended for flock screening [50,86]. However, variations in specificity and sensitivity are known for *Mycoplasma* species and need to be considered. Such variations were mainly reported in contexts with the application of inactivated vaccines against different other bacterial agents [260–264]. Thus, reactors have to be confirmed by HI and suspected flocks are generally investigated with molecular methods as well. Over the last century, successful control of *Salmonella* Gallinarum/Pullorum has been achieved by applying RSA in the field [50,265]. Furthermore, a recent study showed the high specificity of this test system even in flocks vaccinated against *Salmonella* Enteritidis which was often seen as a problem due to cross-reactions [266].

Different to this, the majority of ELISA systems are used for vaccination control or even to predict the time point of vaccination as shown for infectious bursal disease [267]. However, ELISA kits from different manufacturers may vary in specificity and/or sensitivity, resulting in conflicting titer profiles [268], which needs to be considered when interpreting results and for decision making. Archived serum samples are helpful to retrospectively trace pathogen transmission as shown for vertically induced adenovirus gizzard erosion (AGE) [15].

So far, single assays are very widespread, with the need to perform individual tests for each pathogen on independent samples. Moreover, the majority of commonly applied serological assays do not allow the differentiation of antigenically related organisms with consequences to differentiate between vaccination response or natural infection. Furthermore, it is not possible to discriminate serological response against a specific serotype for which more cumbersome test systems, such as hemagglutination inhibition or neutralization assays, have to be applied. Recent developments based upon recombinant antigens and proteins have improved assay specificity and facilitated standardization of interpretation. Recent developments enable the differentiation between birds being vaccinated with chemically inactivated viruses or with viral vector vaccines and birds being infected with field virus, as shown for, e.g., avian influenza virus, NDV, ILTV, IBDV and FAdV [269–272].

Recently, microsphere immunoassays were developed, enabling the measurement and discrimination of numerous analytes with the power of automation. Such bead-based systems were developed to detect antibodies against multiple pathogens or subtypes simultaneously as shown for influenza A virus, NDV, IBV, ILTV, FAdV and/or aHEV [273–278]. With this, much more data are obtained in comparison to a single component. The large number of analytes to be measured offers high potential and can combine several pathways of host response. This might not only include immunological response characterized by antibodies, but can also include broader physiological parameters to assess the well-being of animals such as heat shock proteins or biomarkers for gut heath. Finally, for pathogen detection, biosensors gained certain attractiveness due to their application in the field [279]. Practicability simply depends on the technical composition of the system. Those systems gained importance especial to detect avian influenza virus on site as reviewed by Astill et al. [19].

## 8. Histology

Microscopic changes in organs and tissue structures caused by infectious agents are important features for diagnosis, even though the occurrence of pathognomonic lesions is exceptional and differences in the anatomy of birds and mammals or diseases affecting particular bird species may cause difficulties in the interpretation of histological changes [280]. However, the exact description of microscopic changes in tissues can be a significant contribution for a presumptive diagnosis and it is even more vital for endemic pathogens with substantial variation in pathogenicity.

For histopathology, different tissue preparations can be processed to evaluate changes in host organs, on the organ or cellular levels. Most commonly, tissue samples are fixed in formalin, dehydrated, paraffin-embedded and stained with hematoxylin-eosin staining [281]. In recent decades, different techniques for the specific detection of molecular structures in cells have been developed [282]. These include tools such as immunohistochemistry and in situ hybridization using specific antibodies and nucleic acid probes. This is of particular relevance for the detection of pathogens that cannot be identified or determined by morphological characteristics. Some diseases, such as Marek's disease, do not always cause typical histological changes and require further diagnostic confirmation. The presumptive diagnosis of Marek's disease relies on identifying the involved tissues (inner organs, eye, skin and nerves) and the main type of involved lymphoid cells (T cells) to discriminate it from lymphoid leucosis (Figure 1) [283].

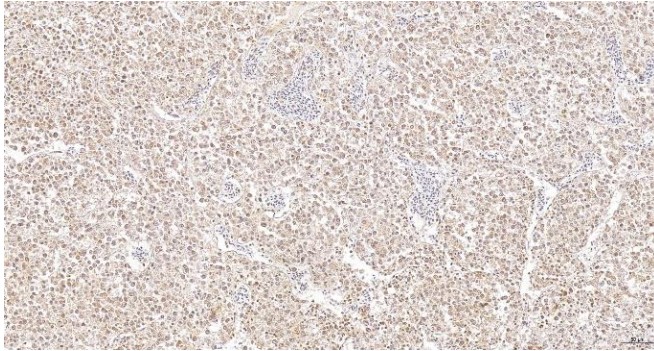

**Figure 1.** Immunohistochemistry for the detection of T cells (brown stained cells) in lymphoma caused by Marek's disease virus.

However, reticuloendotheliosis can transform T cells, potentially infiltrating nerve tissue [284,285]. Furthermore, peripheral neuropathy results in similar lesions compared to Type B lesions in nerves reported for Marek's disease (Figure 2) [286].

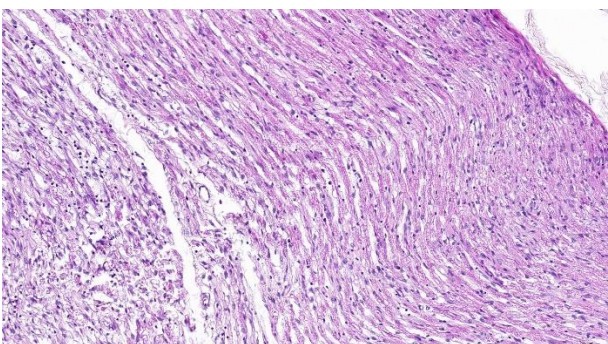

**Figure 2.** Diffuse infiltration of pleomorphic lymphoid cells mainly indicating peripheral neuropathy, Marek's disease or reticuloendotheliosis.

Therefore, the presumptive diagnosis of Marek's disease must be confirmed by the detection of the virus in the tissue with the Meq antigen being constantly expressed in MDV tumors [287]. Consequently, recent advances using histological tools for the detection of MDV involve the application of polyclonal and monoclonal antibodies against Meq that can be used for the specific diagnosis of the disease in paraffin embedded tissues [288,289].

For other viral diseases, inclusion bodies in host cells might be pathognomonic in histopathological findings due to their specific characteristics, such as the location in an infected cell (nucleus, cytoplasm) as well as the staining affinity using HE. Based on such parameters, Kato et al. [290] differentiated intra-cytoplasmatic accumulations in Type A and Type B inclusions, both observed in cells infected with fowlpox. Similarly, intranuclear hepatic inclusions in the course of viral hepatitis must be well examined for a valid presumptive diagnosis as the causative virus may be tentatively determined by the structure of inclusion bodies [291]. In a recent case of hepatitis in pheasants, histopathology clearly revealed inclusion bodies [241]. Inclusion body hepatitis (IBH) caused by certain fowl adenoviruses (FAdVs) of the genus *Aviadenovirus* is not described in pheasants, although such birds can suffer from marble spleen disease (MSDV), a member of the genus *Siadenovirus*, characterized by inclusions within mononuclear cells of the spleen [292,293]. However, based on the histopathological changes and the features of the intranuclear inclusions, showing amphophilic to acidophilic prominent inclusion bodies that filled the nucleoplasm of hepatocytes (Figure 3a), a parvovirus was suggested to be the etiological agent. The use of specific probes against the parvovirus DNA confirmed the presence of this virus in the nuclei of cells (Figure 3b).

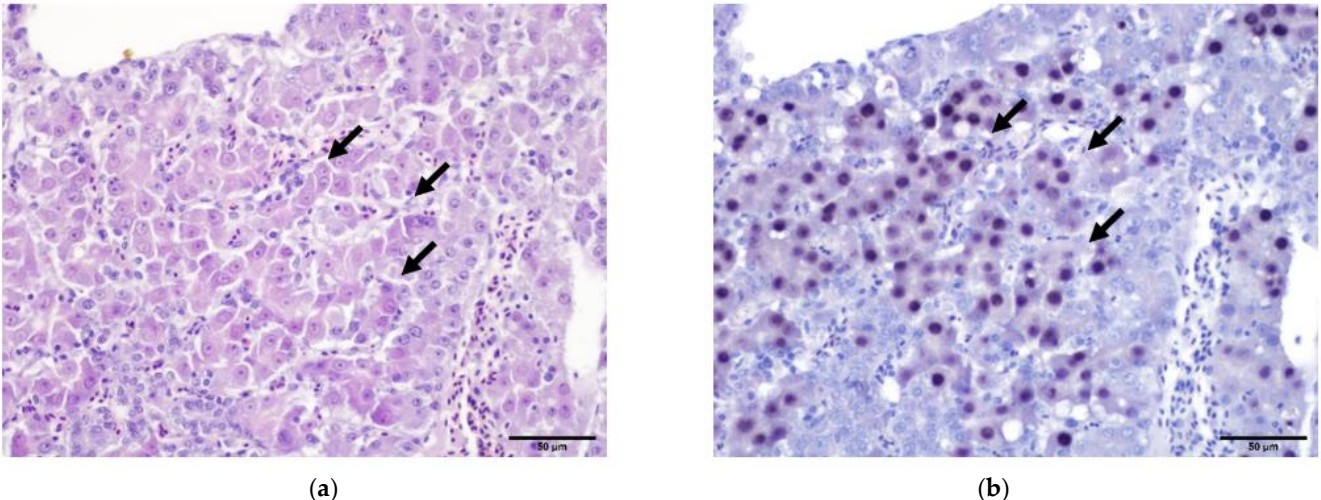

(**a**)                                                                                                                                 (**b**)

**Figure 3.** (**a**) Intranuclear inclusions indicating viral hepatitis in the liver of a pheasant (arrow). (**b**) Specific identification of a parvovirus by in situ hybridization (arrow).

Advanced histological techniques are also crucial in diagnosing pathogens which are microscopically visible. The protozoan *Histomonas meleagridis* is a parasite of poultry that infiltrates different organs and can cause high morbidity and mortality in infected flocks [10]. The parasite is observed in tissue samples stained by hematoxylin-eosin (HE) or the Periodic acid-Schiff method [294,295] but low numbers in tissues without inflammation can easily be missed. To increase the sensitivity of conventional staining protocols, in situ hybridization and immunohistochemistry were developed [296,297]. By this, a single parasite can be detected and differentiation to other flagellates is possible. Furthermore, *Histomonas meleagridis* causes similar lesions in liver and ceca of poultry such as other trichomonads [298]. Conventional staining reveals the presence of the protozoa (Figure 4a) but in situ hybridization allows the differentiation of *Histomonas meleagridis* and *Tetratrichomonas gallinarum* (Figure 4b) [299]. This can furthermore be necessary for the diagnosis of other infectious agents such as bacteria to localize and identify the relevant pathogenic species of co-infected birds as previously reported [300].

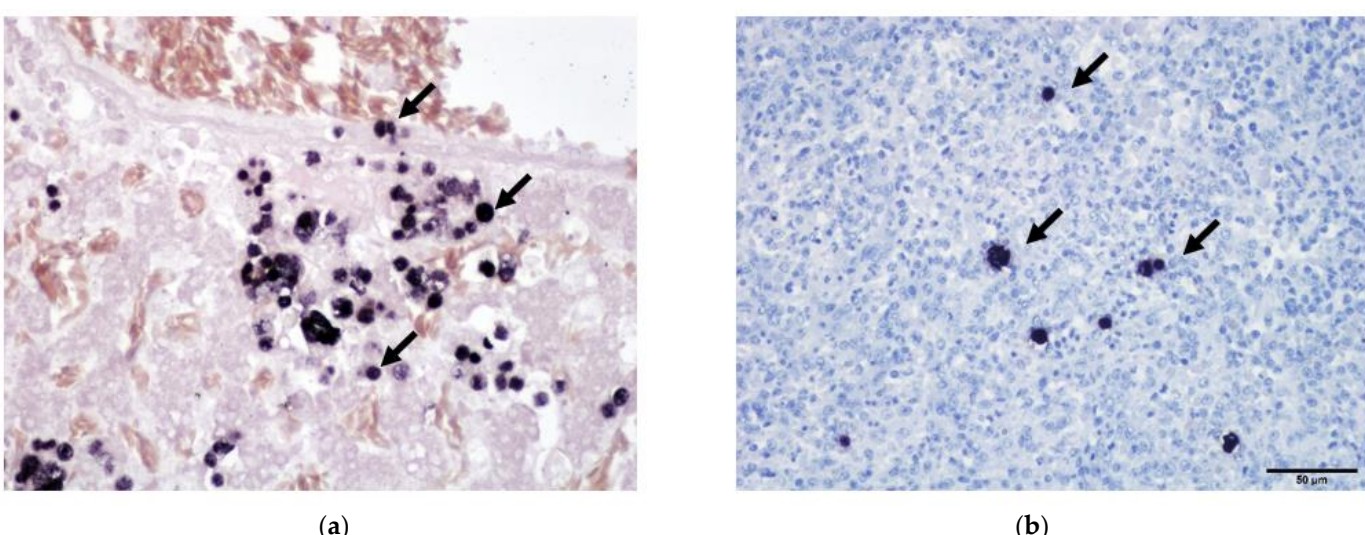

(**a**)                                                                                                                                 (**b**)

**Figure 4.** In situ hybridization for the specific detection and localization of (**a**) *Histomonas meleagridis* and (**b**) *Tetratrichomonas gallinarum* (arrow).

In the future, digitalization of histological preparations is expected to be more and more implemented [301]. This implies automatic detection of tissue structures such as

different chicken leukocytes in digital images of histological sections and its quantification without human bias as recently demonstrated [302]. Furthermore, computerized quantification can be combined with the localization of a pathogen and specific immune cells in the same tissue to identify relevant immune traits [303]. Digital pictures of microscopic lesions are useful for documentation of cases and can be stored in an archive and data repositories. By this, pictures with metadata can be made accessible worldwide to be used for different studies and applications. Computer-aided diagnosis can support the examiner in evaluating microscopic changes. The use of artificial intelligence (AI) enables the development of algorithms for specific histopathological changes. AI is already being much used in the medical field, including infectious disease diagnostics [304]. However, there are still certain limitations of AI-based computer-assisted pathological diagnosis, including a lack of confidence in diagnostic results, inconvenience in practical use and simplicity of function. [305]. Nevertheless, it can be expected that digitalization of histopathology and the application of AI is a promising approach that will improve diagnosing poultry diseases.

## 9. Outlook and Challenges

The sustainability of commercial poultry production depends on quick and accurate detection of impaired flock health in order to set up interventions and to adjust prophylactic strategies. Considering animal welfare, targeted treatment and efficacious prophylaxis, together with the extent and quality of diagnostic procedures, have to be constantly improved keeping in mind that the lack of efficacious drugs and chemicals is a serious limitation. Standardization of laboratory techniques is an important aspect of disease diagnosis and modern laboratories possess an externally validated quality assurance system. This includes numerous features, one of which is the regular participation at interlaboratory comparisons to evaluate the performance of a certain procedure. We already demonstrated the value of such comparisons for accurate detection of *Mycoplasma gallispeticum* and *Mycoplasma synoviae* by PCR approximately 20 years ago [306]. In the field of PCR or serology these comparisons are very common today, but they remain exceptional in disciplines such as histology, where personal experience and skills dominate the technology, a characteristic of bioinformatics as well. This highlights the importance of linking expertise with technologies to develop a holistic approach as a basic strategy to diagnose (infectious) diseases in poultry (Figure 5).

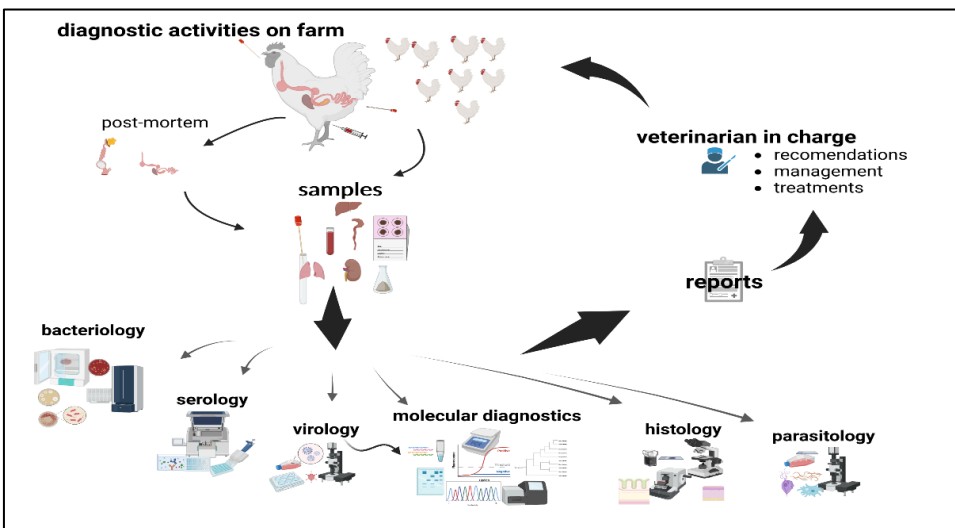

**Figure 5.** Diagram to demonstrate the holistic approach for diagnosing infectious poultry diseases (created with BioRender.com).

In this concept, diagnosing poultry diseases starts at the farm with precise recoding and detailed anamnesis. To date, PLF tools for early detection and recognition of

diseases on farm are not routinely available, as most data have been acquired under experimental conditions and farm equipment has to be updated. Nevertheless, it can be expected that with additional research, leading to progress in technology, the validation of behavioral tracking and analyses of flock data collected in poultry house environments, PLF developments offer great potential for the future of disease diagnostics in poultry. In the laboratory, diagnostic methods will be further developed and bioinformatic tools will become more and more important, replacing conventional methods, with molecular techniques versus pathogen isolation being at the forefront. This will coincide with more demanding equipment and technologies, although numerous conventional methods will remain, and pathogen isolation would be a good example to study functional aspects but also to produce certain types of vaccines.

Consequently, in the future, more information technology will be applied to connect diagnostic investigations with the farm environment. Modern laboratories already use a database accessible via a mobile app to trace samples during investigations and to follow up records. This indicates the need for close interaction between all people involved in diagnostic investigations keeping in mind that specialization is ongoing, and skills of experts will diverge even more. In such a scenario, it remains challenging to interlink people and infrastructure in order to generate a surplus as outlined in Figure 5. This is more easily established in integrated production systems with the downside that data are less accessible for research and science.

**Author Contributions:** Conceptualization, introduction and conclusions, D.L., I.B., B.G., C.H. and M.H.; diagnostic activities on farm and serology, B.G.; bacteriology, C.H.; parasitology and histology, D.L.; molecular diagnostics, I.B.; virology and outlook, I.B. and M.H. All authors have read and agreed to the published version of the manuscript.

**Funding:** This research received no external funding.

**Institutional Review Board Statement:** Not applicable.

**Informed Consent Statement:** Not applicable.

**Data Availability Statement:** Not applicable.

**Acknowledgments:** Figure 5 was created with BioRender.

**Conflicts of Interest:** The authors declare no conflict of interest.

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
