# Peer review of "Diagnosing Infectious Diseases in Poultry Requires a Holistic Approach: A Review"

_poultry, doi:10.3390/poultry2020020_

Round 1

Reviewer 1 Report

This is a well written and interesting review paper. Minor changes are recommended to correct spelling, grammar and punctuation errors. Moreover, sources should be added in figures. 

Author Response

The authors thank the reviewer for the positive response. The reviewer did not raise any specific comments to be addressed.  The manuscript underwent  substantial editing on grammar and style with correction of language and typos.  An updated version of the manuscript applying track and change mode is attached. 

Reviewer 2 Report

This is an excellent comprehensive review of diagnostic methods used to detect infectious diseases in poultry caused by different organisms. It is well-written and easy to understand and comprehend. Only a few minor edits need to be made as outlined below.

1. The first sentence in the abstract is a bit vague or perhaps a comma should be added after "described" for clarity.

2. Line 85: replace the semicolon with a parenthesis)

3. Line 96: Better move the figure to make it Figure 1 since it's the first mentioned in the manuscript and because it's more suited here as an introduction to the diagnostic methods.

4. Line 153: Change "Enterobacteriaceae" to "Enterobacterales" as the latter follows the new nomenclature.

5. Lines 159, 165, 173, 174, 175, 185: Spell out the genus of all the bacteria if this is their first occurrence in the manuscript.

6. Line 196: Better cite the latest versions of EUCAST and CLSI of 2023.

7. Line 203: Spell out NGS/WGS.

8. Line 268: Spell out mini-FLOTAC.

9. Line 389: Change "considers" to "consider"

10. Line 485: You can just use the abbreviation if the spell out will be provided on line 203 as in comment #8 above.

Author Response

The authors thank the reviewer for the positive response on our review. The manuscript underwent substantial editing on grammar and style with correction of language and typos. An updated version of the manuscript applying track and change mode is attached. 

Specific comments:

1. The first sentence in the abstract is a bit vague or perhaps a comma should be added after "described" for clarity.

- the first sentence was changed

2. Line 85: replace the semicolon with a parenthesis)

- done as suggested

3. Line 96: Better move the figure to make it Figure 1 since it's the first mentioned in the manuscript and because it's more suited here as an introduction to the diagnostic methods.

- We thought about this but we think the figure should be kept in the "Outlook and Challenges" as it nicely summarized the main message.  This would be less suitable for a specific chapter as it remains applicable toall of them. However, the sentence in line 96 (now 104) is rephrased to avoid that "Figure 5" is mentioned at the beginnig.  

4. Line 153: Change "Enterobacteriaceae" to "Enterobacterales" as the latter follows the new nomenclature.

- changed as suggested

5. Lines 159, 165, 173, 174, 175, 185: Spell out the genus of all the bacteria if this is their first occurrence in the manuscript.

- changed and adapted as suggested

6. Line 196: Better cite the latest versions of EUCAST and CLSI of 2023.

- changed as suggested

7. Line 203: Spell out NGS/WGS.

- changed as suggeted

8. Line 268: Spell out mini-FLOTAC.

- changed as suggested

9. Line 389: Change "considers" to "consider"

- changed as suggested

10. Line 485: You can just use the abbreviation if the spell out will be provided on line 203 as in comment #8 above.

- changed as suggested

Reviewer 3 Report

Review:

 Diagnosing infectious diseases in poultry asks for a holistic approach: a review.

 General comments:

The publication presents a review of the diagnostic methods applied to avian pathology with a holistic approach.

The paper reports in the various chapters all the methodologies applied in diagnostics of avian diseases, both the classic ones and the more modern ones including the PLF.

The work appears complete, full of practical as well as scientific ideas.

Each chapter is extensively developed with relevant and current bibliographic entries.

The figures and tables are clear and pertinent to the topics covered.

Author Response

The authors thank the reviewer for the positive response on our review. The manuscript underwent substantial editing on grammar and style with correction of language and typos. This should address the comment given by the reviewer about the language and style. An updated version of the manuscript applying track and change mode is attached. 

Reviewer 4 Report

The review is interesting since it integrates the different diagnostic techniques of infectious diseases in birds. However, it would be appropriate to summarize the text in the form of figures, tables, etc. to make the content more attractive to the reader.

In addition, it is important to review the wording of the entire document and consider all the punctuation marks necessary to make the sentences understandable, as well as the rewriting of some sentences.

In the attached file you can see in greater detail the suggestions made to improve the quality of the review.

Author Response

The authors thank the reviewer for the positive response on our review. The comments and suggestions given in the attached file were all considered and the manuscript was adapted accordingly. Furthermore, the manuscript underwent substantial editing on grammar and style with correction of language and typos. An updated version of the manuscript applying track and change mode is attached. 

Reviewer 5 Report

The manuscript is very interesting but the authors should clear mode of action entire the cell using figures. Also the authors encouraged to specify the poultry species in table 1 contents. In addition the authors should revised the English editing to be sure that there are no grammatical errors.

Author Response

The authors thank the reviewer for the positive response on our review. An updated version of the manuscript applying track and change mode is attached. 

Specific comments: 

The manuscript is very interesting but the authors should clear mode of action - entire the cell using figures.

- We do think that the mode of action is addressed in Figure 5. Economic figures can not be given for each individual subject, this would be beyond the review and the skills of the authors. 

Also the authors encouraged to specify the poultry species in table 1 contents.

- All studies listed in Table 1 were done in chickens, except a single one. This is now addressed in a legend of the table. 

In addition the authors should revised the English editing to be sure that there are no grammatical errors.

- The manuscript underwent substantial editing on grammar and style with correction of language and typos.

Round 2

Reviewer 4 Report

In the attached manuscript a couple of grammatical observations are presented, which are highlighted in yellow.
